# Effect of Graphene on Flame Retardancy of Graphite Doped Intumescent Flame Retardant (IFR) Coatings: Synergy or Antagonism

**Yachao Wang \***  **and Jiangping Zhao**

Department of Safety Engineering, College of Resource Engineering, Xi'an University of Architecture & Technology, Xi'an 710055, China; zhaojiangping@xauat.edu.cn

\* Correspondence: wangyachao@xauat.edu.cn; Tel.: +86-29-8220-5869

**Abstract:** A comparative study between graphene and modified graphene oxide (mGO) on the flame retardancy of graphite doped intumescent flame retardant (IFR) coatings is preliminarily investigated by cone calorimeter (CC), XRD, and SEM, with the final aim of clarifying the interactions between different graphenes and graphite doped coatings (polyester resin-ammonium polyphosphate-urea-pentaerythritol). The CC results determine that graphene exerts an obviously antagonistic effect on flame resistance, evidenced by the increased peak heat release rate (p-HRR) of 56.9 kW·m$^{-2}$ for $S_{D8+graphene}$ (sample coating contains graphite with a particle size of 8 μm and 0.5 wt.% graphene as dopant), which increased by 80.6% compared with $S_{D8}$ (coating contains graphite with a particle size of 8 μm); substitution with graphene or mGO imparts an acceleration of fire growth, because graphene inertness improves the viscosity of melting system, evidenced by the cracked appearance and porous structure of $S_{D8+graphene}$. However, the higher reactivity of mGO favors the combustion; the barrier effect inhibits the transfer of mass and heat simultaneously, leading to a slight influence on flame retarding efficiency.

**Keywords:** graphene; intumescent flame retardant; antagonistic effect; heat release rate; cone calorimeter

## 1. Introduction

Flame retardancy of polymeric materials is a subject of major concern due to the need to minimize fire risk and meet fire safety requirements [1], especially for timber structures (plywood, density board, and fiberboard), which are widely used as interiorly decorative materials. Construction materials have long been required to resist burn through or sudden loss of mechanical properties, and maintain structural integrity, whilst continuing to stay intact when exposed to fire or heat [2,3]. Consequently, surface coating with flame retardant paints or varnishes emerges as an available, effective, and efficient surface-treatment method. Intumescent flame retardant (IFR) coatings have been employed as ecologically-friendly substitutes for halogen-containing flame retardant additives [4], which favor the formation of shielding char layers, providing adequate fire suppression and/or thermal insulation [2,3], due to their facile, cost-effective, and energy-efficient synthesis. IFR coatings can be used not only for wood fire protection, but also (especially) for thermal insulation protection of steel, which prevents the loss of mechanical properties in case of fires, and allows for more effective evacuation and fire extinguishing [3].

However, more recently, properties of coatings that need to be functionalized and modified by hybrid nano-materials are attracting increasing attention. Because of the possibility of simultaneous improvement of several coating parameters [5–8], the inorganic substance combination

of nano-particles with other conventional flame retardant-containing systems appears as a promising way to find a substrate with superior flame retardancy [9]. Furthermore, expensive and complicated organic-inorganic hybrid nano-materials have been developed, not only for fire protection improvement in coatings, but also for simultaneous enhancement of several of its properties.

Additionally, hybrid materials combining graphene or modified graphene oxide (mGO) with flame retarding elements have aroused growing interest [10,11]. It is reported that the dosage of graphene at very low loadings (0.2–0.7 wt.%) can provide significant reinforcement to materials [9–12] used to design novel flame retardants. Graphene can promote the formation of compact, dense, and uniform chars in the condensed phase during the combustion of polymer matrices; the incorporation of graphene can provide a so-called "tortuous path" effect that significantly alters the diffusion path of pyrolysis products, thus resulting in significantly reduced mass loss rate [13]. The network, consisting of carbon nanotubes and graphene nano-sheets, plays a very important role in enhancing intumescent char during combustion of nano-composites [14].

However, the following major technical shortcomings restrict the use of bare graphene as a flame retardant: (1) Homogeneous dispersion of graphene in the polymer matrix, and (2) relatively lower efficiency as a flame retardant, when graphene is used alone [15,16]. Although the selection of a suitable polymer binder allows for easy dispersion of carbon fillers in liquid coating compositions via widely used sonication [17] or simple mechanical mixing [18], the narrow choice of intumescent binders for fire protection does not help find a compatible polymer matrix for condensed aromatic structures of carbon particles. Consequently, the layered double hydroxides [19], polydimethylsiloxane (PDMS) [20,21], and polysiloxane [22] have been employed as dispersants to prepare graphene-modified hybrids, which enhance flame resistance. Meanwhile, much effort has been dedicated to the chemical functionalization of graphene or mGO with the objective of improving its solubility and compatibility with polymers by changing its surface property. This could also enhance the flame resistance of composites, resulting in a restacking-inhibited and porous three-dimensional structure composed of ultrathin nano-petals [23–25].

Furthermore, there is scarce literature on flame retardancy of graphene or mGO modified IFR coatings, and further exploration on graphene-doped IFR is necessary to enrich the practical applications of graphene and mGO. This paper compares graphene and mGO and investigates the flame retardancy of graphite-doped IFR coatings. In order to guarantee the homogeneous dispersion and excellent water resistance of hybrid coatings, PDMS and sodium silicate are used as assistants; our previous investigation showed a strong compatibility between them using polyester resin (PR) as the binder [26]. Consequently, ammonium polyphosphate (APP), pentaerythrite (PER), urea, and PR were used as the starting materials to prepare a hybrid coating. The effect of graphene and mGO on the flame retardancy of graphite-doped APP–PER–PR-based coatings is preliminarily studied; it aims to clarify the synergistic or antagonistic effect in interactions between them, using plywood as a carrier. Cone calorimeter (CC) is employed to examine the heat release rate (HRR), and the microstructures of char residues are evaluated by X-ray diffraction (XRD) and scanning electron microscope (SEM), respectively.

## 2. Materials and Methods

### 2.1. Raw Materials

$Na_2SiO_3 \cdot 9H_2O$ (analytically pure) with a molar $Na_2O/SiO_2$ rate of 1 was purchased from Tianjin Fuchen chemical reagent company (Tianjin, China). The starting materials for IFR coating consisting of analytically pure PR, APP, urea, and PER, were all purchased from Shanghai Chemical Reagent Company (Shanghai, China). The water-based PR was synthesized from neopentyl glycol, trimethylolpropane (TMP), 1,4-cyclohexanedimethanol,1,6-hexanediol, adipic acid, dipropylene glycol butyl ether (DPNB), and dimethylolpropionic acid (DMPA), using mono-*N*-butyltin oxide (MBTO) as the catalyst, as well as an antifoaming agent and other additives through melting, followed by

azeotropic distillation, which was fabricated by Baihong environmental protection technology company of Dongguan in Guangdong province. Its solid content, viscosity, acid number, and hydroxyl number was 70%, 6800 mPa·s, 40 mg KOH g$^{-1}$, and 90 mg KOH g$^{-1}$, respectively. The viscosity of PR and sample coatings at 25 °C was tested by the NDJ-5T viscosimeter of Fangrui instruments in Shanghai, China, with a precision of 0.1 mPa·s, according to DIN 53019 standard [26]. Plywood was purchased from a timber processing plant in Xi'an with second-class flame retardancy. The two kinds of spherical graphite were purchased from Tengsheng Chemical Group (Qingdao, China); the graphite with an average particle size of 8 μm was denoted as D8, and D18 represented graphite with an average particle size of 18 μm, respectively. The graphene and mGO were obtained from XianFeng Chemical Group in Nanjing of Jiangsu province in China. The specific surface of graphene is about 1600 m$^2$·g$^{-1}$ with a thickness of 0.8–1.2 nm and diameter of 0.5–5 μm, while that of mGO powder is about 780 m$^2$·g$^{-1}$ with a thickness of 0.2–5 μm. The specific surface of D8 is about 7.5 m$^2$·g$^{-1}$, while that of D18 is about 5.0 m$^2$·g$^{-1}$. Figure 1 presents that the morphology obviously varies for different graphite; the uniformly sized particles with blunt and smooth surface are observed in Figure 1a, and the alveolate plates appear in Figure 1b, indicating the large internal space to accommodate or absorb small molecules. However, the irregular particles with rough surfaces and quite different sizes are observed in Figure 1c, revealing a high reactivity in theory.

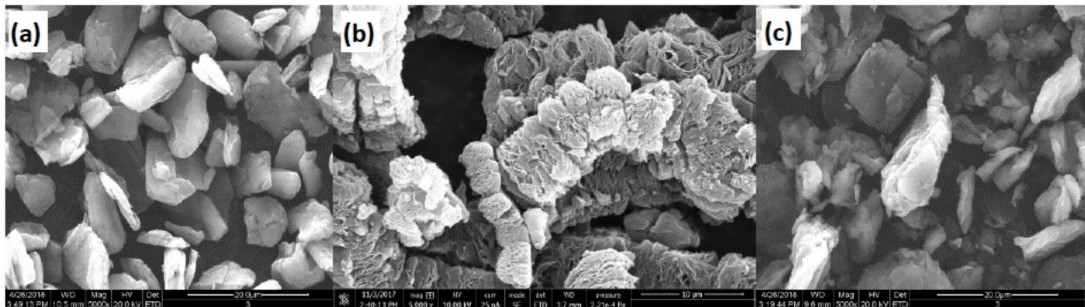

**Figure 1.** SEM of starting carbonaceous materials with 5000× zoom including (**a**) D8, (**b**) graphene, and (**c**) mGO.

## 2.2. Preparation of Samples

Firstly, the IFR coating was prepared by slowly dispersing Na$_2$SiO$_3$·9H$_2$O in distilled water; 0.2 g Na$_2$SiO$_3$·9H$_2$O was directly dispersed into 30 mL water with a constant temperature of 60 °C under a stirring rate of 60 rpm on the magnetic stirrer, which lasted about 5 min to obtain a uniform solution. It was used to neutralize the PR for obtaining a water-soluble coating, and also provided some –Si≡OH or –Si–O–Si– short chains. Then the 4 g PER, 7 g APP, 4 g urea, 8 g PR, and 0.27 g PDMS were added sequentially at 60 °C over 5 min, maintaining a continuous stirring of 200 r·min$^{-1}$ for approximately 10 min, respectively. The graphite was added into the aforementioned APP–PER–PR-based coating over 5 min by vigorous stirring of 1000 r·min$^{-1}$ for approximately 15 min to form a homogeneous binder; water evaporation was prohibited during the process [27]. The total content of spherical graphite powder was 2.7 g with a weight percent of 5 wt.%, which consisted of D8, D18, graphene, and mGO as shown in Table 1, and the graphite was replaced by graphene or mGO with a weight percent of 0.5 wt.%. Finally, the binder was manually brushed on to the surface of plywood with a thickness of 0.5 mm tested by the film thickness gauge (PosiTector 200, DeFelsko, Ogdensburg, NY, USA). The angle between the brusher and the plywood surface was 55–65°, the surface drying time and drying time were 38 min and 2.4 h, respectively, and the amount of coating was approximately 500 g·m$^{-2}$. The samples covered by the hybrid coatings were called S$_{D8}$, S$_{D8+graphene}$, S$_{D18}$, S$_{D18+graphene}$, S$_{D8+mGO}$, and S$_{D18+mGO}$, respectively. The viscosity was 64, 124, 47, 89, 76, and 53 mPa·s for S$_{D8}$, S$_{D8+graphene}$, S$_{D18}$, S$_{D18+graphene}$, S$_{D8+mGO}$, and S$_{D18+mGO}$, respectively, which was tested by an NDJ-5T viscosimeter at 25 °C when preparation of the coating sample was complete.

**Table 1.** Compositions of organic-inorganic hybrid coatings.

| Samples | APP/g | PER/g | Urea/g | PR/g | $H_2O$/g | PDMS/g | $Na_2SiO_3 \cdot 9H_2O$/g | Graphite/g |
|---|---|---|---|---|---|---|---|---|
| $S_{D8}$ | 7 | 4 | 4 | 8 | 30 | 0.27 | 0.2 | 2.7(D8) |
| $S_{D8+graphene}$ | 7 | 4 | 4 | 8 | 30 | 0.27 | 0.2 | 2.43(D8) + 0.27(Graphene) |
| $S_{D18}$ | 7 | 4 | 4 | 8 | 30 | 0.27 | 0.2 | 2.7(D18) |
| $S_{D18+graphene}$ | 7 | 4 | 4 | 8 | 30 | 0.27 | 0.2 | 2.43(D18) + 0.27(Graphene) |
| $S_{D8+mGO}$ | 7 | 4 | 4 | 8 | 30 | 0.27 | 0.2 | 2.43(D8) + 0.27(mGO) |
| $S_{D18+mGO}$ | 7 | 4 | 4 | 8 | 30 | 0.27 | 0.2 | 2.43(D18) + 0.27(mGO) |

## 2.3. Characterizations

Heat release property was detected by a cone calorimeter (CC, ZY6243, Zhongnuo Instrument Company, Dongguan, China) according to ISO-5660-1-2015 [27]. The sample wrapped in an aluminium foil with a size of $95 \times 95 \times 3$ mm$^3$ was exposed horizontally to an external heat flux of 35 kW·m$^{-2}$ (600 °C approximately) to catch the developing fire [25]. The following parameters were obtained, which included time to ignite (TTI) and time of flaming combustion ($T_f$) corresponding to the time from TTI to extinguishment of flame, which directly was observed by the naked eye. The peak heat release rate (p-HRR) and time to p-HRR ($T_p$) were recorded. Total heat release (THR) was the accumulated heat release during the whole combustion. The fire performance index (FPI) was defined as the ratio of TTI and p-HRR as FPI = TTI/p-HRR, and the fire growth index (FGI) was defined as the ratio of p-HRR and $T_p$ as FGI = p-HRR/$T_p$. The average effective heat of combustion (AEHC) was defined as the ratio of THR and weight loss (WL) as AEHC = THR/WL.

Micro-morphologies of samples after burning were observed on Quanta 200 SEM (FEI, Hillsboro, OR, USA), equipped with an energy dispersive spectrometer (EDS, FEI, Hillsboro, OR, USA) under condition of 20 kV. XRD patterns of samples were recorded by the D/MAX-2400 X-ray diffractometer (Rigaku, Tokyo, Japan), equipped with a rotation anode using Cu Kα radiation.

## 3. Results

### 3.1. Combustion Characteristics

Figure 2 presents distinct increases in the p-HRR for samples with graphene, regardless of graphene or mGO, compared with that of a sample with graphite only. It shows that substitution with graphene leads to a remarkably increased p-HRR, which increases from 31.5 kW·m$^{-2}$ ($S_{D8}$) to 56.9 kW·m$^{-2}$ for $S_{D8+graphene}$, while that of $S_{D18}$ climbs from 36.1 to 57.2 kW·m$^{-2}$ ($S_{D18+graphene}$), accompanied by the retardant emergence of $T_p$ compared to that of the sample without graphene, thus ascribing to the physical shielding effect of graphene. Meanwhile, the samples with mGO present slight increases in p-HRR in comparison to that of the corresponding samples, indicating the absence of a synergistic effect on flame retardancy between graphene/mGO and graphite doped APP–PER–PR-based coatings, but an obvious antagonistic effect in the sample with graphene. It determines that the replacement of spherical graphite by graphene is detrimental in obtaining enhanced flame retardancy of APP–PER–PR-based coatings.

Table 2 briefly lists the HRR parameters of the samples. The TTI increases for the samples with graphene, while it decreases for the sample with mGO, compared with the corresponding samples with neat spherical graphite, indicating a higher reactivity of mGO in APP–PER–PR-based coatings during combustion. Sonnier et al. [28] suggested that the decrease of TTI with the incorporation of multiwall carbon nanotubes (MWCNT) was due to more absorbed heat at the surface. Additionally, our result demonstrates that substitution with graphene or mGO causes an acceleration of FGI to the graphite doped APP–PER–PR-based coatings, and exhibits decreases in the FPIs. However, it is worth pointing out that the $T_f$ decreases for samples with graphene or mGO, and the AEHC of samples with mGO declines, when compared to the corresponding samples with graphite only, which might be contributed to the reinforcement of char. However, that of the samples with graphene increases, revealing that graphene imparts an accelerated flaming combustion, to a certain extent.

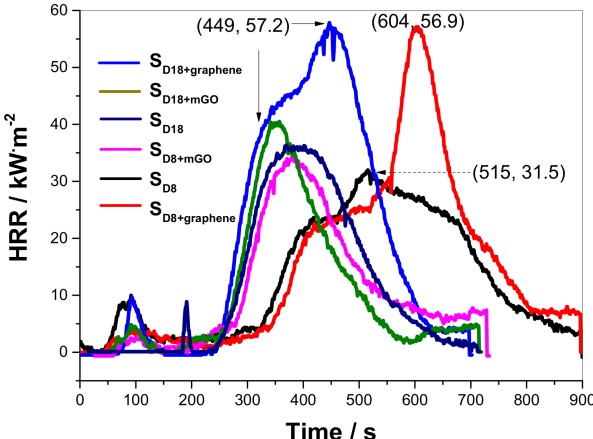

**Figure 2.** HRR of sample recorded by CC.

**Table 2.** HRR parameters of samples tested by CC.

| Samples | TTI/s | $T_f$/s | $T_p$/s | p-HRR/ kW·m$^{-2}$ | FPI/ s·m$^2$·kW$^{-1}$ | FGI/ kW·m$^{-2}$·s$^{-1}$ | WL$_b$/g | THR/ MW·m$^{-2}$ | AEHC/ kW·g$^{-1}$ |
|---|---|---|---|---|---|---|---|---|---|
| S$_{D8}$ | 288 ± 3 | 122 ± 6 | 515 ± 8 | 31.5 ± 2 | 9.14 | 0.06 | 20.5 ± 0.2 | 10.36 ± 0.5 | 4.45 |
| S$_{D8+graphene}$ | 318 ± 3 | 59 ± 6 | 604 ± 10 | 56.9 ± 4 | 5.59 | 0.09 | 21.6 ± 0.3 | 10.97 ± 0.5 | 4.47 |
| S$_{D18}$ | 259 ± 3 | 143 ± 6 | 387 ± 8 | 36.1 ± 2 | 7.17 | 0.09 | 18.9 ± 0.2 | 7.87 ± 0.4 | 3.66 |
| S$_{D18+graphene}$ | 296 ± 3 | 78 ± 6 | 449 ± 9 | 57.2 ± 4 | 4.67 | 0.13 | 20.2 ± 0.2 | 13.51 ± 0.6 | 5.89 |
| S$_{D8+mGO}$ | 267 ± 3 | 87 ± 6 | 379 ± 8 | 33.9 ± 2 | 4.07 | 0.09 | 19.2 ± 0.2 | 7.96 ± 0.4 | 3.65 |
| S$_{D18+mGO}$ | 138 ± 3 | 110 ± 6 | 351 ± 9 | 40.1 ± 3 | 3.44 | 0.11 | 19.8 ± 0.2 | 7.11 ± 0.4 | 3.16 |

Figure 3 presents an increase in smoke temperature for samples with graphene, while decreases in smoke temperature are detected for samples with mGO, compared with the corresponding samples without graphene or mGO. It is well known that the faster the combustion, the higher the smoke temperature is, implying that doped graphene facilitates fire propagation. This provides a strong validation of the HRR result, indicating that substitution with graphene in D8 doped APP–PER–PR-based coatings is detrimental to improving its flame resistance.

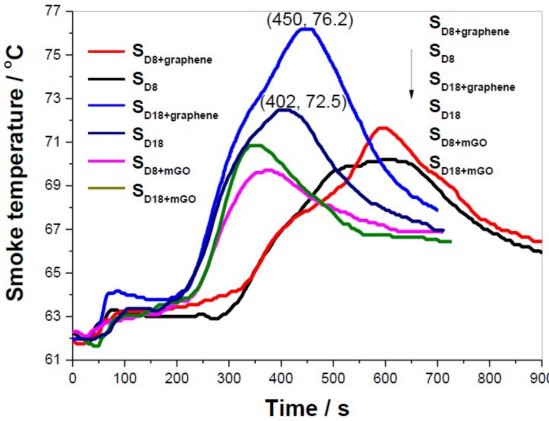

**Figure 3.** Smoke temperature of exhaust pipe during combustion of samples.

Figure 4 presents similar changing rules compared to the results of HRR—the graphene triggers a distinct increase in oxygen consumption, leading to a dramatic decrease in O$_2$ concentration, implying that strong combustion occurs for samples with graphene. However, the samples with mGO presents a slight decrease in O$_2$ concentration compared with that of the corresponding samples with neat D8 or D18. It also proves that the added graphene accelerates the combustion of hybrid coatings.

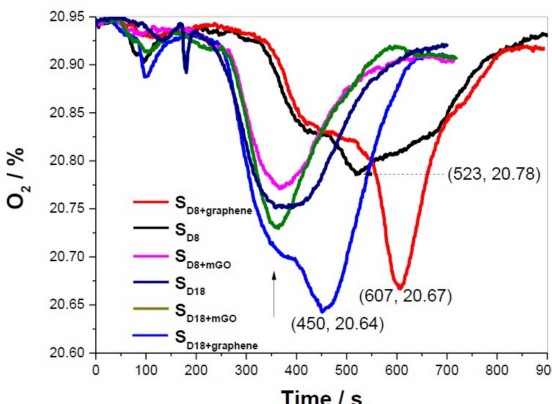

**Figure 4.** $O_2$ concentration of exhaust pipe during combustion of samples.

It is of consensus that the $CO_2$ yield is proportional to oxygen consumption during burning of combustible organic material. The highest $CO_2$ concentration of 0.40% is for $S_{D18+graphene}$ (Figure 5), while $S_{D8}$ exhibits the lowest $CO_2$ concentration. This is consistent with results of HRR and $O_2$ concentration, determining that graphene generates an antagonistic effect with the graphite doped APP–PER–PR-based coatings, taking flame resistance into account.

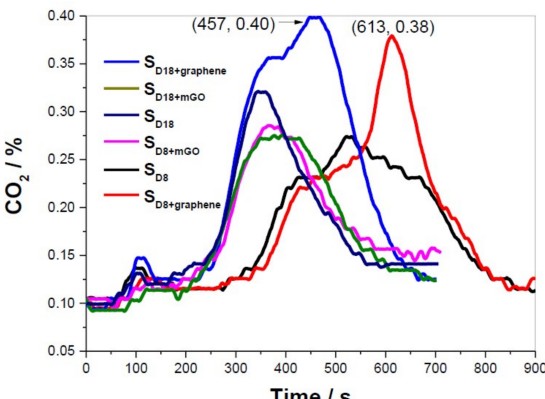

**Figure 5.** $CO_2$ concentration of exhaust pipe during combustion of samples.

### 3.2. XRD of Residual Char

The pattern of residual char presents a hump at $2\theta = 15–40°$ in Figure 6a, which is assigned to the amorphous carbonaceous materials derived from combustion of $S_{D8}$. Compared to that of raw D8, the sharp peak corresponding to graphite covers the weak hump. It reveals the coexistence of graphite and exfoliated carbon in the residual char. This occurred because, in the case of the layered nanofillers, the shift of the diffraction pattern to the lower values of $2\theta$, and its blur, confirm increment of the interlayer spacing [29]. Meanwhile, a sharp peak corresponding to graphite superimposed on the amorphous hump in 15–40° ($2\theta$) is obviously observed in Figure 6b, although the sharp peak at ($2\theta = 11.5°$) covers the hump for mGO. Because mGO is exfoliated into individual graphene sheets, it causes the regular and periodic structures of graphene to disappear [14]. Similarly, the exfoliated graphene is covered by a sharp peak of graphite derived from D8 in Figure 6c, but the hump in 15–40° ($2\theta$) is obviously observed for residual char. The weak and nearly flat pattern of raw graphene indirectly reveals an inherent inertness of graphene.

It is worth noting that the sharper the peak, the larger the graphite particle size is, as shown in Figure 6d, indicating a higher crystalline and lower reactivity, leading to a decrease in flame retardancy. A pattern of weak intensity assigned to amorphous carbonaceous substance is recorded in Figure 6d,

which derives from the sample without graphite. It provides positive evidence that the doped graphite covers the hump of residual char.

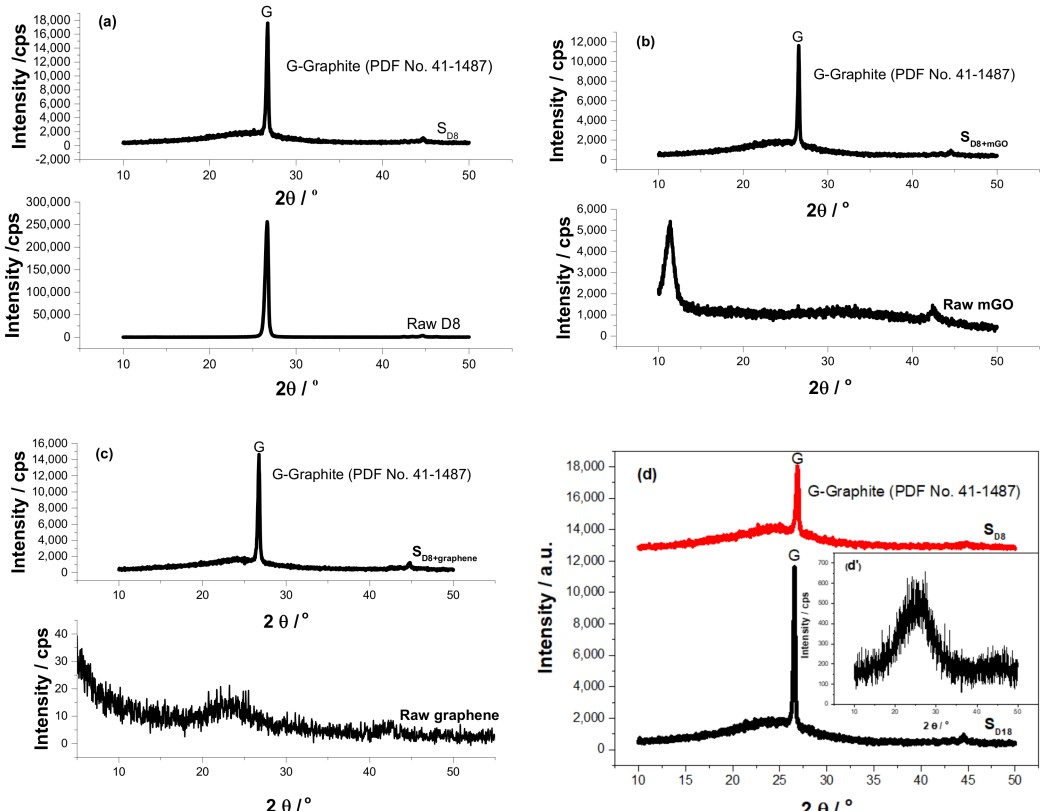

**Figure 6.** XRD of residual char, including: (**a**) $S_{D8}$, (**b**) $S_{D8+mGO}$, (**c**) $S_{D8+graphene}$, (**d**) comparison between SD8 and $S_{D18}$, and (**d'**) sample without graphite.

### 3.3. Morphology of Residual Char

Figure 7 presents the appearances of char residue. The obvious phenomena of melting-solidifying process are pronounced for $S_{D8}$ in Figure 7a, and several homogeneous bubbles disperse on the surface of the char residue layer. As for $S_{D18}$, the bubbles are absent and plate-like chars with many cracks cover the surface in Figure 7b. However, a thinner char layer generates for $S_{D8+graphene}$; the char is prone to cracking and uncases the underlying plywood in Figure 7c. Especially for $S_{D18+graphene}$, the char layer peels off due to its loose structure and poor bonding, leading to the appearance of underlying plywood with large cracks in Figure 7d, which might be due to incomplete esterification caused by the filling and barrier effect of graphene. On the contrary, the samples with mGO exhibit slight changes compared with that of samples without mGO. A swollen surface with lots of bubbles is observed in Figure 7e; more and smaller bubbles are generated on the surface with many cracks, as shown in Figure 7f.

Figure 8 presents a laminar char layer with thickness of approximately 2 μm for $S_{D8}$, which is stacked disorderly with a rough surface in Figure 8a. The char layer grows with higher continuity compared with that of $S_{D8}$, when graphene is added to the IFR coating; however, a loose and porous structure is observed in Figure 8b, indicating that the graphene reversely facilitates the compactness, leading to a broken residual char layer. However, a smooth, irregular, and continuous sheet with a thickness of 2–3 μm appears for $S_{D8+mGO}$ in Figure 8c, which completely covers the underlying substance to protect effectively from the fire, corresponding to enhanced flame resistance, compared to $S_{D8+graphene}$.

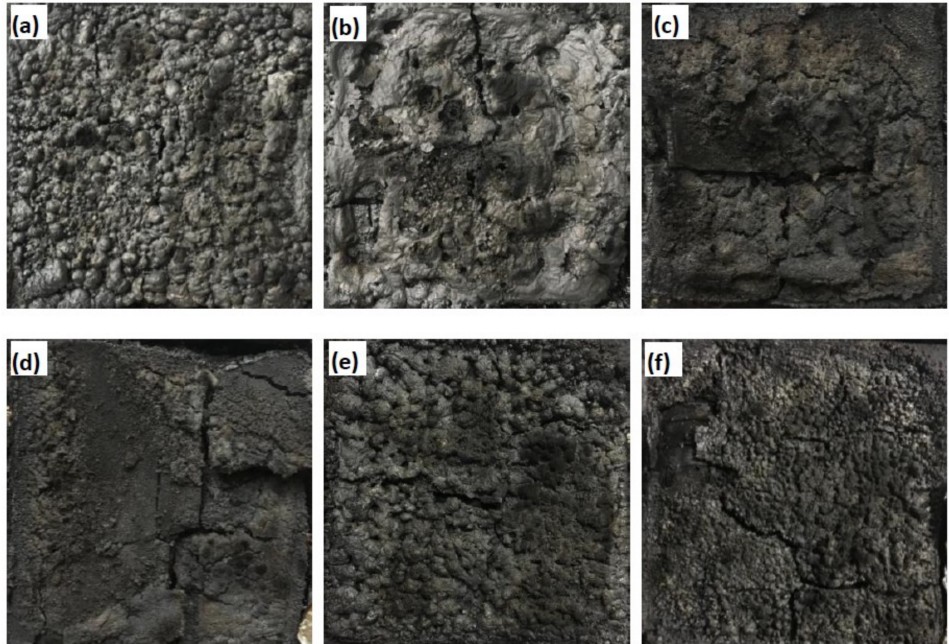

**Figure 7.** Digital photo of samples after burning in CC, including: (**a**) S$_{D8}$, (**b**) S$_{D18}$, (**c**) S$_{D8+graphene}$, (**d**) S$_{D18+graphene}$, (**e**) S$_{D8+mGO}$, (**f**) S$_{D18+mGO}$ (1:2).

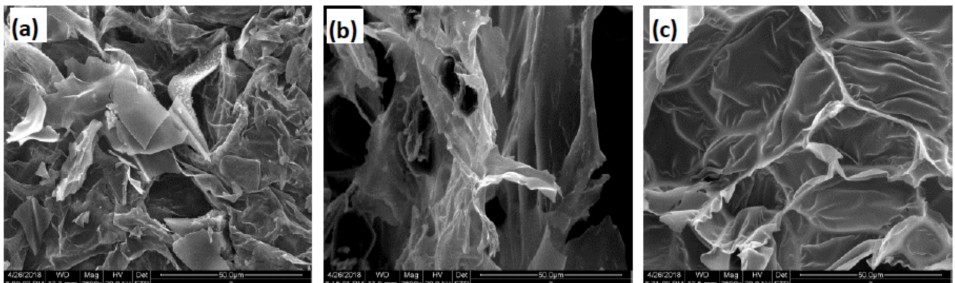

**Figure 8.** SEM of residual char with 2500× zoom, including: (**a**) S$_{D8}$, (**b**) S$_{D8+graphene}$, and (**c**) S$_{D8+mGO}$.

## 4. Discussions

APP–PER–PR-based coatings generate: (1) melting and decomposition of PER (about 25%) during 200–300 °C, (2) APPs degrade into phosphoric acid between 300 and 400 °C, subsequent esterification between phosphoric acid and PER occurs; accompanied by the release of NH$_3$, H$_2$O is derived from urea and yields swollen charred layers, (3) further charring and degradation of APP at about 600 °C, and carbonaceous char starts to decompose above 500 °C, and (4) the coating gelatinizes and solidifies to form a porous carbon foam layer when the reaction is nearly complete [30].

The nano-sheet graphene is accessible for molecular adsorption due to the large surface area. The delocalized π-electron system of graphene provides strong affinity for molecules with aromatic rings [31]. Furthermore, the incorporation of graphene can provide a so-called "tortuous path" effect that significantly alters the diffusion path of pyrolysis products, thus resulting in significantly reduced mass loss rate [13,32], leading to distinctly delayed emergency of $T_p$. The delayed $T_p$ for samples with graphene is contributed to the following reasons: (1) The high thermal conductivity facilitates the absorption of heat from radiating cone in CC, and the physical shielding effect hinders heat diffusion, leading to the slow rise of APP–PER–PR-based coatings. (2) The layered labyrinth effect retards the paths of flame propagation, together with the spherical graphite.

However, the initial melting of PER and APP is of importance in the subsequent esterification, and the melt viscosity of IFR coating plays a crucial and essential role in the effective and efficient formation

of viscous swollen char. However, the fine particles and inherent chemical inertness of graphene favor the dramatic increase in the viscosity of the melting system, simultaneously [33], due to its strong π-π force, weak van der Walls force [34], and poor compatibility with the IFR coating. Moreover, APP could generate polyphosphoric acid and increase the melt viscosity at a higher temperature [35], leading to an incomplete esterification, evidenced by the thinner char layer and cracked char residue. The spherical graphite has been used as a lubricant to favor flowing property of molten coatings, but the excessive viscosity mainly traps and suppresses the diffusion of foaming gases including $NH_3$ and $H_2O$; thus, the formation and development of viscous swollen char is inhibited. Generally, the doped graphite is used as a carburant in APP–PER–PR-based coatings; the spherical shape is beneficial to improving the viscosity of the coating during heating, but the porous and exfoliated lamellar graphene in the melt favors dramatic increase in viscosity. It speculates a balance between the viscosity, and the rate of expansion foaming during the combustion is crucial to form a continuous and compact shielding layer. Moreover, graphene is prepared using a chemical exfoliation route, which is scalable and of low cost [36]; there inevitably is a large number of defects in the graphene nano-sheet due to changes in its aromatic structure and formation of oxygen-containing groups [37,38]. Graphene is also contaminated with alkali metal salt by-products left from GO reduction synthesis due to increment of the interlayer spacing, leading to a significant decrease in its thermal stability [39]. Exfoliated graphene with numerous defects and alkali metal salt is detrimental in enhancing the flame retardancy of graphite-containing IFR coatings. Because most exfoliated graphite fillers are derived from graphene intercalation compounds, the layered graphite is intercalated with atoms or molecules, such as alkali metals or mineral acids, which increases its interlayer spacing, weakening interlayer interactions, and facilitating the exfoliation [39,40]. However, the formation of a continuous and compact shielding layer is crucial in improving the flame resistance of IFR coating, and a compatibility between the coating matrix and graphene is necessary to constitute a synergistic flame-retarding effect. Consequently, the chemically modified graphene has huge potential in developing novel composites with diverse functionality.

On the other hand, substitution with mGO in graphite-doped APP–PER–PR-based coatings exhibits a small influence on the flame retardancy, because mGO has higher reactivity, ascribed to abundant oxygen-containing groups on its surface, which provide more active sites for covalent attachment and non-covalent adsorption [41,42]. The oxygen moieties can also be easily removed by heating or a variety of reagents [43], leading to decreases in TTI and $T_p$. However, the flame retardancy produces a slight change, due to its barrier or blocking effect, which reinforces the amorphous char derived from graphite doped APP–PER–PR-based coatings during combustion. In other words, the competition between barrier and charring should be the major flame retardant mechanisms of the graphite doped APP–PER–PR-based coatings. The barrier contains physical blocking properties for transfer of heat or mass due to incombustibility; the charring depends on the initial esterification between APP and PER, which is also affected by mGO due to reactivity and a filling effect.

It was reported that MWCNT is negligible or insignificant in intumescent coating, because the anti-oxidation and thermal stability properties of coatings with and without MWCNT remained the same at later stages of the experiment due to degradation of MWCNTs at high temperature. The residual weights of coatings without MWCNT and with a concentration of 0.5 wt.% were almost the same [44]. However, our investigation suggests that a remarkably antagonistic effect exists between graphene and graphite doped APP–PER–PR-based coatings. It demonstrates that the mGO exerts a negligible influence on the flame retardancy of IFR coating, and the excessive barrier of carbonaceous substance in IFR coating is detrimental to fireproof performance. However, the interface characteristics involved in graphite doped APP–PER–PR-based coatings need to be investigated further in the future, and real-time supervision of the viscosity of melt during the formation of the char layer is necessary and essential for illustrating the char-forming mechanism.

## 5. Conclusions

A comparative study between graphene and mGO on the flame retardancy of graphite doped IFR coatings is preliminarily investigated in this paper, with the aim of clarifying the interactions between different graphenes and graphite doped APP–PER–PR-based coatings. The following conclusions are drawn.

- The CC results determine that there is an obviously antagonistic effect on flame retardancy between graphene and graphite doped APP–PER–PR-based coatings, evidenced by the increased p-HRR of 56.9 kW·m$^{-2}$ for $S_{D8+graphene}$ (increased by 80.6% compared with $S_{D8}$). Substitution with graphene or mGO imparts an acceleration of FGI to the graphite doped APP–PER–PR-based coatings.
- The competition between barrier and charring should be the main flame retardant mechanisms of the graphite doped APP–PER–PR-based coatings. The coexistence of graphite and amorphous carbon for residual char is detected from the XRD results. However, the poor compatibility and inherent inertness of graphene improved the viscosity of the melting system, evidenced by a thinner and cracked appearance for $S_{D8+graphene}$, presenting a loose and porous structure.

**Author Contributions:** Conceptualization, Y.W. and J.Z.; Methodology, Y.W.; Software, Y.W.; Validation, Y.W.; Formal Analysis, Y.W.; Investigation, Y.W.; Resources, J.Z.; Data Curation, J.Z.; Writing—Original Draft Preparation, Y.W.; Writing—Review and Editing, Y.W.; Visualization, Y.W.; Supervision, J.Z.; Project Administration, Y.W.; Funding Acquisition, Y.W.

**Funding:** This research was funded by the Shanxi National Science Foundation (No. 2017JQ5066), Shaanxi Provincial Education Department Fund (17JK0447), and China Scholarship Council (CSC No. 201808610034).

**Conflicts of Interest:** The authors declare no conflict of interest.

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
