# Peer review of "Effect of Graphene on Flame Retardancy of Graphite Doped Intumescent Flame Retardant (IFR) Coatings: Synergy or Antagonism"

_coatings, doi:10.3390/coatings9020094_

Round 1
Reviewer 1 Report
In order to evaluate the manuscript, it should be better that the authors make some modifications, as indicated in the document.

Author Response
Thanks for your carefully and accurately reviewing and kindly guidance, the following is our response to your questions and suggestions, which help us to improve the understanding of flame retardant coating, writing skills, and other professional knowledge.
Review 1 #
1. In order to evaluate the manuscript, it should be better that the authors make some modifications, as indicated in the document.
Thank you for kindly reminding, the superfluous words have been deleted.
2. The color of curves is missing
Thank you for your advice, we have revised the Fig.2~5 carefully with color short lines.
3. the wrong word of
Thanks for carefully reviewing, we are sorry for the carelessness, it has been revised.
4. the graph with poor quality
Thank you for your advice, we have revised the graph carefully with smoothed curve and good quality.
5. Does it make sense the characterization of the residual char by XRD?
Thank you for your question, we are sorry to give the little explaination on the phenomenon. The XRD patterns are used to prove the coexistence of graphite and exfoliated carbon in the residual char. It happened, because in case of layered nanofillers the shift of the diffraction pattern to the lower values of 2 Theta angle, and its blur, can confirm the increment of the interlayer spacing, which provides the positive evidence for the reduced flame resistance, and it has been illustrated in the revised manuscript. Generally, it proposes that the exfoliated graphene with numerous defect and alkali metal salt is detrimental to enhance the flame retardancy of graphite-containing IFR coatings.
Apart from the abovementioned revision, we also try our best to check the sentences and grammar, consulting the latest literatures and polishing the article. We would like to express our great appreciation to you and reviewers for comments on our paper, we obtain in-depth understanding on composite coating, and our English language skills are improved. Looking forward to hearing from you.

Reviewer 2 Report
Article on flame-retardant intumescent coatings testing the influence of different carbon fillers on flame retardancy of polyester coatings applied on plywood substrate. Curiously, an antagonistic effect on this parameter between graphene and graphite has been observed. In my opinion, this interesting discovery deserves to be published in “Coatings” after taking into account the following comments, given in chronological order:
1.Lines 15-18: Sample shortcuts in Abstract have no explanation. Please, correct these, eg. “coating with graphite and graphene” instead of “SD8+graphene”.
2. L. 16-18: Please, correct the sentence: “Because the inertness of graphene facilitates the enhanced viscosity of melting system…” to make it grammatically correct.
3. L. 25-29: The sentence is too long. It can be finished in L. 28: “…fiberboard. Timber product…”
4. L. 24-38: It have to be added, that intumescent coatings can be used not only for wood fire protection, but also (especially) for thermal insulation protection of steel, which prevents the loss of its mechanical properties in case of fire, which allow more effective evacuation and fire extinguishing. Some words have to be also specified according to polymer coating terminology.
It can be done by an extension of L.28-33 sentence:
“Construction materials have long been required to resist burn through or sudden loss of mechanical properties and maintain structural integrity whilst continuing to provide when exposed to fire or heat [2] [2a]. Consequently, surface coating with flame retardant paints or varnishes emerges as an available, effective, and efficient surface-treatment method, and the intumescent flame retardant (IFR) coatings have been employed as ecologically-friendly substitutes for halogen-containing flame retardant additives [3], which favor the formation of shielding char layers providing adequate fire suppression and/or thermal insulation [2] [2a], due to their facile, cost-effective, and energy-efficient synthesis.
[2a] E.D. Weil, Fire-protective and flame-retardant coatings - A state-of-the-art review, J. Fire Sci. 29 (2011) 259–296. doi:10.1177/0734904110395469.
5. L. 34-38. It have to be indicated, that rather expensive and complicated organic-inorganic hybrid nanomaterials have been developed not only for coatings fire protection improvement, but for enhancement of many coatings properties, simultaneously. It can be done by an extension of L.34-36:
“But more recently their properties to be functionalized and modified by hybrid nano-materials have attracted increasing attentions, because of the possibility of simultaneous improvement of several coating parameters [3a, 4-6].”
[3a] S. Kugler, K. Kowalczyk, T. Spychaj, Influence of dielectric nanoparticles addition on electroconductivity and other properties of carbon nanotubes-based acrylic coatings, Prog. Org. Coat. 92 (2016) 66–72. doi:10.1016/j.porgcoat.2015.12.006.
6. L. 49-51: The problem with homogeneous dispersion in polymers affects all carbon fillers: graphene, GO, graphite, carbon nanotubes etc. and, first of all, its most simple solution is the use of polymer binders with affinity for carbon structures, which should be underlined here. Moreover, the price of carbon nanostructures can be problematic too, but, fortunately, this is not a technical problem. I recommend to deal with these as follows:
“However, following major technical shortcomings restrict the use of bare graphene as a flame retardant: (1) homogeneous dispersion of graphene in the polymer matrix, and (2) relatively lower efficiency as a flame retardant, when graphene is used alone [13-14]. Although the selection of suitable polymer binder allows for easy dispersion of carbon fillers in liquid coating compositions via widely used sonication [14a] or simple mechanical mixing [14b], the narrow choice of intumescent binders for fire protection does not help to find compatible polymer matrix for condensed aromatic structures of carbon particles. Consequently, the layered double hydroxides [15], polydimethylsiloxane…”
[14a] F. Wu, W. Zhao, H. Chen, Z. Zeng, X. Wu, Q. Xue, Interfacial structure and tribological behaviours of epoxy resin coating reinforced with graphene and graphene oxide, Surf. Interface Anal. 49 (2017) 85–92. doi:10.1002/sia.6062.
[14b] S. Kugler, K. Kowalczyk, T. Spychaj, Transparent epoxy coatings with improved electrical, barrier and thermal features made of mechanically dispersed carbon nanotubes, Prog. Org. Coat. 111 (2017) 196–201. doi:10.1016/j.porgcoat.2017.05.017.
7. L. 76: The following parameters of polyester resin (PR) have to be added: (1) trade name, trade symbol (if it is commercially available); (2) chemical formula and monomers used in synthesis of the resin; (3) viscosity (mPas); (4) manufacturer – NOT supplier (“Shanghai chemical reagent company” is a too general term for a serious scientific journal).
8. L. 78-81: Specific surface in m^2/g of all carbon fillers (graphites, graphene, mGO) as well as thickness of mGO and graphene has to be given.
9. L. 90: Dispersing method of Na2SiO3 has to be given.
10. L. 90-103: Viscosity of all coating compositions (SD8, SD8+graphene, SD18, SD18+graphene, SD8+mGO, and SD18+mGO) has to be given.
11. L. 134: Please, add colors to the legend of Fig.2. They can be color short lines or color fonts.
12. L. 140: Please, add MWCNT shortcut here (or earlier, in L. 46)
13. L. 146: Please, correct “graphenen”
14. L. 155, 163, 170: Please, add colors to legends as in Fig.2
15. L. 173-183: It is worth to explain a little bit more the phenomenon we can observe for polymeric materials with layered fillers via XRD technique: I recommend to add the following sentence:
“It reveals the coexistence of graphite and amorphous or exfoliated carbon in the residual char. It happened, because in case of layered nanofillers the shift of the diffraction pattern to the lower values of 2Theta angle, and its blur, can confirm the increment of the interlayer spacing [23a]. Meanwhile, the sharp peak corresponding to…”
[23a] S. Kugler, T. Spychaj, K. Wilpiszewska, K. Goracy, Z. Lendzion-Bielun, Starch-graft copolymers of N-vinylformamide and acrylamide modified with montmorillonite manufactured by reactive extrusion, J. Appl. Polym. Sci. 127 (2013) 2847–2854. doi:10.1002/app.37630.
16. L. 174-188, 253, 279: Please, consider the use of the word “amorphous”. The most of graphene nanostructures have more than one carbon layer, so they are still crystalline, although graphene powder is surely more amorphous than graphite. Maybe “exfoliated” would be a better word? Maybe you should define the boundary between crystallinity and amorphousness? Please, discuss these questions.
17. L. 258-261: Please, ensure that MWCNT shortcut was previously defined.
Author Response
Dear editor and reviewers,
Thanks for your carefully and accurately reviewing and kindly guidance, the following is our response to your questions and suggestions, which help us to improve the understanding of flame retardant coating, writing skills, and other professional knowledge.
Review 2 #
Article on flame-retardant intumescent coatings testing the influence of different carbon fillers on flame retardancy of polyester coatings applied on plywood substrate. Curiously, an antagonistic effect on this parameter between graphene and graphite has been observed. In my opinion, this interesting discovery deserves to be published in “Coatings” after taking into account the following comments, given in chronological order:
1. Lines 15-18: Sample shortcuts in Abstract have no explanation. Please, correct these, eg. “coating with graphite and graphene” instead of “SD8+graphene”.
Thanks for carefully reviewing, we are sorry for the carelessness, it has been revised. “SD8+graphene (sample coating contains graphite with a particle size of 8 μm and 0.5 wt% graphene as dopant), which increased by 80.6% compared with SD8 (coating contains graphite with a particle size of 8 μm)”
2. L. 16-18: Please, correct the sentence: “Because the inertness of graphene facilitates the enhanced viscosity of melting system…” to make it grammatically correct.
Thank you for carefully reviewing, we are sorry for the carelessness, it have been revised
3. L. 25-29: The sentence is too long. It can be finished in L. 28: “…fiberboard. Timber product…”
Thank you for kindly reminding, it has been divided into short sentences.
4. L. 24-38: It have to be added, that intumescent coatings can be used not only for wood fire protection, but also (especially) for thermal insulation protection of steel, which prevents the loss of its mechanical properties in case of fire, which allow more effective evacuation and fire extinguishing. Some words have to be also specified according to polymer coating terminology.
It can be done by an extension of L.28-33 sentence:
“Construction materials have long been required to resist burn through or sudden loss of mechanical properties and maintain structural integrity whilst continuing to provide when exposed to fire or heat [2] [2a]. Consequently, surface coating with flame retardant paints or varnishes emerges as an available, effective, and efficient surface-treatment method, and the intumescent flame retardant (IFR) coatings have been employed as ecologically-friendly substitutes for halogen-containing flame retardant additives [3], which favor the formation of shielding char layers providing adequate fire suppression and/or thermal insulation [2] [2a], due to their facile, cost-effective, and energy-efficient synthesis.
[2a] E.D. Weil, Fire-protective and flame-retardant coatings - A state-of-the-art review, J. Fire Sci. 29 (2011) 259–296. doi:10.1177/0734904110395469.
Thank you for kindly suggestion, it is very necessary, which makes the expression precise and coherent.
5. L. 34-38. It have to be indicated, that rather expensive and complicated organic-inorganic hybrid nanomaterials have been developed not only for coatings fire protection improvement, but for enhancement of many coatings properties, simultaneously. It can be done by an extension of L.34-36:
“But more recently their properties to be functionalized and modified by hybrid nano-materials have attracted increasing attentions, because of the possibility of simultaneous improvement of several coating parameters [3a, 4-6].”
[3a] S. Kugler, K. Kowalczyk, T. Spychaj, Influence of dielectric nanoparticles addition on electroconductivity and other properties of carbon nanotubes-based acrylic coatings, Prog. Org. Coat. 92 (2016) 66–72. doi:10.1016/j.porgcoat.2015.12.006.
Thank you for warm-hearted corrections, we are careless to omit this important literature, and we have carefully read this paper, which provides a novel idea to design functionalized coating.
6. L. 49-51: The problem with homogeneous dispersion in polymers affects all carbon fillers: graphene, GO, graphite, carbon nanotubes etc. and, first of all, its most simple solution is the use of polymer binders with affinity for carbon structures, which should be underlined here. Moreover, the price of carbon nanostructures can be problematic too, but, fortunately, this is not a technical problem. I recommend to deal with these as follows:
“However, following major technical shortcomings restrict the use of bare graphene as a flame retardant: (1) homogeneous dispersion of graphene in the polymer matrix, and (2) relatively lower efficiency as a flame retardant, when graphene is used alone [13-14]. Although the selection of suitable polymer binder allows for easy dispersion of carbon fillers in liquid coating compositions via widely used sonication [14a] or simple mechanical mixing [14b], the narrow choice of intumescent binders for fire protection does not help to find compatible polymer matrix for condensed aromatic structures of carbon particles. Consequently, the layered double hydroxides [15], polydimethylsiloxane…”
[14a] F. Wu, W. Zhao, H. Chen, Z. Zeng, X. Wu, Q. Xue, Interfacial structure and tribological behaviours of epoxy resin coating reinforced with graphene and graphene oxide, Surf. Interface Anal. 49 (2017) 85–92. doi:10.1002/sia.6062.
[14b] S. Kugler, K. Kowalczyk, T. Spychaj, Transparent epoxy coatings with improved electrical, barrier and thermal features made of mechanically dispersed carbon nanotubes, Prog. Org. Coat. 111 (2017) 196–201. doi:10.1016/j.porgcoat.2017.05.017.
Thank you for warm-hearted corrections, the corrections are necessary to give comprehensive overview on this field.
7. L. 76: The following parameters of polyester resin (PR) have to be added: (1) trade name, trade symbol (if it is commercially available); (2) chemical formula and monomers used in synthesis of the resin; (3) viscosity (mPas); (4) manufacturer – NOT supplier (“Shanghai chemical reagent company” is a too general term for a serious scientific journal).
Thank you for advice. The water-based polyester resin is synthesized from neopentyl glycol, trimethylolpropane(TMP), 1,4-cyclohexanedimethanol,1,6-hexanediol, adipic acid, dipropylene glycol butyl ether (DPNB), and dimethylolpropionic acid (DMPA), using mono-n-butyltin oxide (MBTO) as the catalyst, as well as antifoaming agent and other additives through melting followed by azeotropic distillation, which is fabricated by Baihong environmental protection technology company of Dongguan in Guangdong province. Its solid content, viscosity, acid number, and hydroxyl number is 70%, 6800 mPa·s, 40 mg KOH·g-1, and 90 mg KOH·g-1, respectively. The viscosity of PR and sample coating at 25 oC is tested by NDJ-5T viscosimeter of Fangrui instruments in Shanghai of China with a precision of 0.1 mPa·s, according to the standard of ISO 3219 [a-b].
[a] Fuensanta, Mónica, Jofre-Reche, José Antonio, Rodríguez-Llansola, Francisco, Costa, Víctor, Iglesias, José Ignacio, & Martín-Martínez, José Miguel. (2017). Structural characterization of polyurethane ureas and waterborne polyurethane urea dispersions made with mixtures of polyester polyol and polycarbonate diol. Progress in Organic Coatings, 112, 141-152.
[b] Patel, A., Patel, C., Patel, M. G., Patel, M., & Dighe, A. (2010). Fatty acid modified polyurethane dispersion for surface coatings: effect of fatty acid content and ionic content. Progress in Organic Coatings, 67(3), 255-263.
8. L. 78-81: Specific surface in m^2/g of all carbon fillers (graphites, graphene, mGO) as well as thickness of mGO and graphene has to be given.
Thank you for advice. The specific surface of graphene is about 1600 m2·g-1 with a thickness of 0.8~1.2 nm and diameter of 0.5~5 μm, while that of mGO powder is about 780 m2·g-1 with diameter of 0.2~5 μm. The specific surface of D8 is about 7.5 m2·g-1, while that of D18 is about 5 m2·g-1.
9. L. 90: Dispersing method of Na2SiO3 has to be given.
Thank you for question. the 0.2 g Na2SiO3·9H2O was directly dispersed into 30 mL water with a constant temperature of 60 oC under a stirring rate of 60 rpm on the magnetic stirrer, which lasted about 5 min to obtain a uniform solution, it was served to neutralize the PR for obtaining the water-soluble coating, and also provided some -Si≡OH or –Si-O-Si- short chains.
10. L. 90-103: Viscosity of all coating compositions (SD8, SD8+graphene, SD18, SD18+graphene, SD8+mGO, and SD18+mGO) has to be given.
Thank you for advice, the viscosity is 64 mPa·s, 124 mPa·s, 47 mPa·s, 89 mPa·s, 76 mPa·s, and 53 mPa·s for SD8, SD8+graphene, SD18, SD18+graphene, SD8+mGO, and SD18+mGO, respectively, which is tested by NDJ-5T viscosimeter at 25 oC when the preparation of coating sample is just completed.
11. L. 134: Please, add colors to the legend of Fig.2. They can be color short lines or color fonts.
Thank you for your advice, we have revised the Fig.2~5 carefully with color short lines.
12. L. 140: Please, add MWCNT shortcut here (or earlier, in L. 46)
Thank you for advice, we have revised it there.
13. L. 146: Please, correct “graphenen”
Thank you for your kindly reminding, we are careless to make this mistake, we are sincerely thank you for your timely correction.
14. L. 155, 163, 170: Please, add colors to legends as in Fig.2
Thank you for your advice, we have revised the Fig.2~5 carefully with color short lines
15. L. 173-183: It is worth to explain a little bit more the phenomenon we can observe for polymeric materials with layered fillers via XRD technique: I recommend to add the following sentence:
“It reveals the coexistence of graphite and amorphous or exfoliated carbon in the residual char. It happened, because in case of layered nanofillers the shift of the diffraction pattern to the lower values of 2Theta angle, and its blur, can confirm the increment of the interlayer spacing [23a]. Meanwhile, the sharp peak corresponding to…”
[23a] S. Kugler, T. Spychaj, K. Wilpiszewska, K. Goracy, Z. Lendzion-Bielun, Starch-graft copolymers of N-vinylformamide and acrylamide modified with montmorillonite manufactured by reactive extrusion, J. Appl. Polym. Sci. 127 (2013) 2847–2854. doi:10.1002/app.37630.
Thank you for your kindly correction, the revised expression is accurate, and the reference is necessary for the explaination in discussion, the increment of the interlayer spacing provides the positive evidence for the reduced flame resistance, we also search some literatures to illustrate the decreased flame resistance in discussion.
16. L. 174-188, 253, 279: Please, consider the use of the word “amorphous”. The most of graphene nanostructures have more than one carbon layer, so they are still crystalline, although graphene powder is surely more amorphous than graphite. Maybe “exfoliated” would be a better word? Maybe you should define the boundary between crystallinity and amorphousness? Please, discuss these questions.
Thank you for your question, the amorphous carbonaceous substance is derived from the combustion of the coating without graphite or graphene, which corresponds to the hump in 15~40o (2θ), as shown in Fig.6d’. Simultaneously, we appreciate your suggestion that using the “exfoliated” to replace the “amorphous”, it makes the expression more accurate to describe the graphene, and we have revised the relevant sentences. However, the boundary between crystallinity and amorphousness involved in graphene is scarce through reviewing literatures.
Furthermore, we add the following representation to elaborate the decreased flame retardancy.
Combining with our experiment, the graphene is prepared from chemical exfoliation route, which is scalable and low cost [1], but it inevitably produces a large number of defects in the graphene nanosheets due to the change in its aromatic structure and the formation of oxygen-containing groups [2-3], and also graphene is contaminated with alkali metal salt by-products left from GO reduction synthesis due to the increment of the interlayer spacing, leading to significant decrease in its thermal stability [4]. Generally, the exfoliated graphene with numerous defect and alkali metal salt is detrimental to enhance the flame retardancy of graphite-containing IFR coating. Because most exfoliated graphite fillers are derived from graphene intercalation compounds, the layered graphite is intercalated by atoms or molecules, such as alkali metals or mineral acids, which increases its interlayer spacing, weakening the interlayer interactions and facilitating the exfoliation [4-5]. However, the formation of continuous and compact shielding layer is crucial for improving the flame resistance of IFR coating, and the compatibility between the coating matrix and graphene is necessary to constitute a synergistic flame-retarding effect. Consequently, the chemically modified graphene has huge potential in developing novel composites with diverse functionality.
[1] Chen, K. , Shi, L. , Zhang, Y. , & Liu, Z. . (2018). Scalable chemical-vapour-deposition growth of three-dimensional graphene materials towards energy-related applications. Chemical Society Reviews, 47, 3018—3036.
[2] Zhang, N. , Zhang, Y. , & Xu, Y. J. . (2012). Recent progress on graphene-based photocatalysts: current status and future perspectives. Nanoscale, 4, 5792–5813.
[3] Jr, K. E. W. , & Sheehan, P. E. . (2014). Graphene synthesis. Diamond and Related Materials, 46(24), 25-34.
[4] Zhu, J. , Chen, M. , He, Q. , Shao, L. , Wei, S. , & Guo, Z. . (2013). An overview of the engineered graphene nanostructures and nanocomposites. RSC Advances, 3(45), 22790.
[5] Potts, J. R., Dreyer, D. R., Bielawski, C. W., & Ruoff, R. S. (2011). Graphene-based polymer nanocomposites. Polymer, 52(1), 5-25.
17. L. 258-261: Please, ensure that MWCNT shortcut was previously defined.
Thank you for your kindly reminding, we are careless to make this mistake, we have revised it in 3.1 of page 8.
Apart from the abovementioned revision, we also try our best to check the sentences and grammar, consulting the latest literatures and polishing the article. We would like to express our great appreciation to you and reviewers for comments on our paper, we obtain in-depth understanding on composite coating, and our English language skills are improved. Looking forward to hearing from you.
Best regard,
Sincerely yours,
Yachao Wang
wangyachao@xauat.edu.cn

Round 2
Reviewer 1 Report
This version of the manuscript can be accept, but, please modify the caption of the Fig.1 adding 5000"x".
Author Response
Answer: Thank you for kindly reminding, we have revised it into “Fig.1 SEM of starting carbonaceous materials with 5000× zoom”, as well as in Fig.8, and the Review 2 also suggested that it needed to correct "with 2500" on "with 2500x zoom", thank you.

Reviewer 2 Report
In my opinion this article can be published in Coatings.
I have found only some minor language mistakes:
1. Line 18: Please, combine into one sentence:
"growth, because the graphene inertness..."
2. Line 43: Please, delete the word "And":
"with superior flame retardancy [9]. Expensive and..."
3. Line 56: Please add "the":
"However, the following major..."
4. Line 117: Please, delete "And":
"short chains. Then the 4g PER, 7g APP..."
5. Line 129: Please, delete "And":
"respectively. The viscosity was 64..."
6. Line 218: Please, delete "And":
"retardancy. The pattern with weak..."
7. Line 244: Please correct "with 2500" on "with 2500x zoom"
Author Response
Answer: Thank you for carefully checking and skilled English, we have revised all the points as you guide, which are marked in revised manuscript, we appreciate your help and suggestions, thanks a lot.
1. Line 18: Please, combine into one sentence:
"growth, because the graphene inertness..."
2. Line 43: Please, delete the word "And":
"with superior flame retardancy [9]. Expensive and..."
3. Line 56: Please add "the":
"However, the following major..."
4. Line 117: Please, delete "And":
"short chains. Then the 4g PER, 7g APP..."
5. Line 129: Please, delete "And":
"respectively. The viscosity was 64..."
6. Line 218: Please, delete "And":
"retardancy. The pattern with weak..."
7. Line 244: Please correct "with 2500" on "with 2500x zoom"
